



# Pseudo-prospective testing of 5-year earthquake forecasts for California using inlabru

Kirsty Bayliss[1], Mark Naylor[1], Farnaz Kamranzad[1], and Ian Main[1]

[1]School of GeoSciences, University of Edinburgh

**Correspondence:** Kirsty Bayliss (Kirsty.Bayliss@ed.ac.uk)

**Abstract.** Probabilistic earthquake forecasts estimate the likelihood of future earthquakes within a specified time-space-magnitude window and are important because they inform planning of hazard mitigation activities on different timescales. The spatial component of such forecasts, expressed as seismicity models, generally rely upon some combination of past event locations and underlying factors which might affect spatial intensity, such as strain rate, fault location and slip rate or past

seismicity. For the first time, we extend previously reported spatial seismicity models, generated using the open source inlabru package, to time-independent earthquake forecasts using California as a case study. The inlabru approach allows the rapid evaluation of point process models which integrate different spatial datasets. We explore how well various candidate forecasts perform compared to observed activity over three contiguous five year time periods using the same training window for the seismicity data. In each case we compare models constructed from both full and declustered earthquake catalogues. In doing

this, we compare the use of synthetic catalogue forecasts to the more widely-used grid-based approach of previous forecast testing experiments. The simulated-catalogue approach uses the full model posteriors to create Bayesian earthquake forecasts. We show that simulated-catalogue based forecasts perform better than the grid-based equivalents due to (a) their ability to capture more uncertainty in the model components and (b) the associated relaxation of the Poisson assumption in testing. We demonstrate that the inlabru models perform well overall over various time periods, and hence that independent data such as

fault slip rates can improve forecasting power on the time scales examined. Together, these findings represent a significant improvement in earthquake forecasting is possible, though this has yet to be tested and proven in true prospective mode.

## 1   Introduction

Probabilistic earthquake forecasts represent our best understanding of the expected occurrence of future seismicity (Jordan and Jones, 2010). Developing demonstrably robust and reliable forecasts is therefore a key goal for seismologists. A key

component of such forecasts, regardless of the timescale in question, is a reliable spatial seismicity model that incorporates as much useful spatial information as possible in order to identify areas at risk. For example in probabilistic seismic hazard modelling (PSHA) a time independent spatial seismicity model is developed by combining a spatial model for the seismic sources with a frequency magnitude distribution. In light of the ever-growing abundance of earthquake data and the presence of spatial information that might help understand patterns of seismicity, Bayliss et al. (2020) developed a spatially-varying point



process model for spatial seismicity using Log-Gaussian Cox processes evaluated with the Bayesian integrated nested Laplace approximation method (Rue et al., 2009) implemented with the open-source R package inlabru (Bachl et al., 2019). Time-independent earthquake forecasts require not only an understanding of spatial seismicity, but also need to prove themselves to be consistent with observed event rates and earthquake magnitudes in the future.

Forecasts can only be considered meaningful if they can be shown to demonstrate a degree of proficiency at describing what future seismicity might look like. The Regional Earthquake Likelihood Model (RELM, Field, 2007) experiment and subsequent Collaboratory for the study of earthquake predictability (CSEP) experiments challenged forecasters to construct earthquake forecasts for California, Italy, New Zealand and Japan (e.g. Schorlemmer et al., 2018; Taroni et al., 2018; Rhoades et al., 2018, and other articles in this special issue) to be tested in prospective mode using a suite of pre-determined statistical tests. The testing experiments found that the best performing model for seismicity in California was the Helmstetter et al. (2007) smoothed seismicity model, whether aftershocks were included or not (Zechar et al., 2013). This model requires no mosaic of seismic source zones to be constructed, requiring only one free parameter - the spatial dimension of the smoothing kernel. In the years since this experiment originally took place, there has been considerable work both to improve the testing protocols and to develop new forecast models which may improve upon the performance of the data-driven Helmstetter et al. (2007) model, primarily by including different types of spatial information to augment what can be inferred from the seismicity alone. Multiplicative hybrid models (Marzocchi et al., 2012; Rhoades et al., 2014, 2015) have shown some promise, but these require some care in construction and further testing is needed. The performance of smoothed seismicity models has been found to be inconsistent in testing outside of California, e.g. with the Italian CSEP experiment finding smoothed past seismicity alone did not do as well as models with much longer term seismicity and fault information (Taroni et al., 2018). Thus, finding and testing new methods of allowing different data types to be easily included in developing a forecast model is an important research goal. Here we explore in particular the role of testing an ensemble of point process simulated catalogues (Savran et al., 2020) in comparison with traditional grid-based tests, where the underlying point process is locally averaged in a grid element.

In this paper we construct and test a series of time-independent forecasts for California by building on the spatial modelling approach described by Bayliss et al. (2020). As a first step in the modelling we take a pseudo-prospective approach to model design, with the forecasts being tested retrospectively on time periods subsequent to the data on which they were originally constructed, and test the models' performance against actual outcome using the pyCSEP package (Savran et al., 2021). This is not a sufficient criterion for evaluating forecast power in true prospective mode, but is a necessary step on the way, and (given similar experience of 'hindcasting' in cognate disciplines such as meteorology) can inform the development of better real-time forecasting models. The results presented here will in due course be updated and tested in true prospective mode, using a training dataset up to the present. We first test the pseudo-prospective seismicity forecasts in a manner consistent with the RELM evaluations. For this comparison we use a grid of event rates and the same training and testing time windows to provide a direct comparison to the forecasts of the smoothed seismicity models of Helmstetter et al. (2007), which use seismicity data alone as an input, and provide a suitable benchmark to our study. We then extend this approach to the updated CSEP evaluations for simulated catalogue forecasts (Savran et al., 2020) and show that the synthetic catalogue-based forecasts perform better





than the grid-based equivalents, due to their ability to capture more uncertainty in the model components and the relaxation of
the Poisson assumption in testing.

## 2   Method

We develop a series of spatial models of seismicity modelled by a time-independent Log-Gaussian Cox Process and fitted with
inlabru, as described in detail in Bayliss et al. (2020), and whose workflow is summarised in Figure 1. The models take as input
twenty years (1984-2004) of California earthquakes with magnitude $\geq 4.95$ from the UCERF3 dataset (Field et al., 2014), with
the magnitude cutoff chosen to be consistent with the RELM forecast criteria. The locations of these events are an intrinsic
component of a point process model with spatially varying intensity $\lambda(\mathbf{s})$, where the intensity is described as a function of
some underlying spatial covariates $x_m(\mathbf{s})$, e.g. input data from seismicity catalogues or geodetic observations of strain rate,
and a Gaussian random field $\zeta(\mathbf{s})$ to account for spatial structure that is not explained by the model covariates. The spatially
varying intensity then can be described with a linear predictor $\eta(\mathbf{s})$ such that

$$\lambda(\mathbf{s}) = e^{\eta(\mathbf{s})}, \tag{1}$$

and $\eta(\mathbf{s})$ can be broken down into a sum of linearly combined components

$$\eta(\mathbf{s}) = \beta_0 + \sum_{m=1}^{M} \beta_m x_m(\mathbf{s}) + \zeta(\mathbf{s}). \tag{2}$$

The $\beta_0$ term is an intercept term, which would describe a spatially homogeneous Poisson intensity if no other components
were included, and each $\beta_m$ describes the weighting of individual spatial components in the model. $\beta_0$ is essentially the uniform
average or base-level intensity, which allows the possibility of earthquakes happening over all of the region of interest as a null
hypothesis, so 'surprises' are possible, though unlikely after adding the other terms and renormalising. The models are built on
a mesh (step 2 of Figure 1) which is required to perform numerical integration in the spatial domain, with the model intensity
evaluated at each mesh vertex as a function of the random field (RF, which is mapped by stochastic partial differential equations
or SPDE in step 3 of Figure 1) and other components of the linear predictor function (equation 2). Fitting the model results in a
posterior probability distribution for each of the model component weights, the random field and the joint posterior probability
distribution for the intensity as a function of these components. The expected number of events can then be approximated by
summing over the mesh and associated weights over the area of interest (Step 5 of figure 1). The performance of the models can
then be evaluated by comparing the expected versus the observed number of events, and the models ranked using the resulting
model deviance information criterion (DIC). DIC is commonly applied in other applications of Bayesian inference, including
inlabru applications to other problems, such as spatial distributions of species in ecology. With the definition used here, DIC is
lower for a model with better predictive skill.

In Bayliss et al. (2020) a range of California spatial forecast models were tested on how well the spatial model created
by inlabru fitted the observed point locations, so were essentially a retrospective test of the spatial model alone in order to







Figure 1. The workflow for generating spatial seismicity models in inlabru, with functions shown on the right.



understand which components were most useful in developing and improving such models. Here we test such models in pseudo-prospective mode for California, again using the approach of testing different combinations of data sets as input data. We develop a series of new spatial models to compare with the smoothed seismicity forecast of Helmstetter et al. (2007). These models contain a combination of four different covariates that were found to perform well in terms of DIC in Bayliss et al. (2020). These are shown in Figure 2 and include the GEM strain rate (Kreemer et al., 2014) (SR) map, NeoKinema model slip rates (NK) attached to mapped faults in the UCERF3 model (Field et al., 2014), a past seismicity model (MS) and a fault-distance map (FD) constructed using the UCERF3 fault geometry, with fault polygons buffered by their recorded dip. The past seismicity model used here is derived from events in the UCERF3 catalogue that occurred prior to 1984. For this data set, we fitted a model which contained only a Gaussian random field to the observed events, thus modelling the seismicity with a random field where we do not have to specify a smoothing kernel, the smoothing is an emergent property of the latent random field. This results in a smoothed seismicity map of events which occurred before our training dataset. This smoothed seismicity model also includes smaller magnitude events and those where the location or magnitude of the event is likely to be uncertain, so may account for some activity that is not observed or explicitly modelled (e.g. due to short-term clustering) at this time. Each of these components (SR, MS, NK, FD) is included as a continuous spatial covariate combined with a random field and intercept component. The M4.95+ events from 1984-2004 are used to construct the point process itself. The exact combination of components in a model is reflected in the model name as set out in Table 1. More details on each of these model components and their performance in describing locations of observed seismicity can be found in Bayliss et al. (2020). Step 7 of the workflow covers the steps described below and results presented here.

## 2.1 Developing full forecasts from spatial models

The inlabru models provide spatial intensity estimates which can be converted to spatial event rates by considering the timescales involved. Since the models we develop here are to be considered time-independent, we assume that the number of events expected in this time period is 'scaleable' in a straight-forward manner, consistent with a (temporally homogeneous) spatially-varying Poisson process. However we know that the rate of observed events is not Poissonian due to observed spatio-temporal clustering (Vere-Jones and Davies, 1966; Gardner and Knopoff, 1974) and that short time-scale spatial clustering can lead to higher rates anticipated in areas where large clusters have previously been recorded (Marzocchi et al., 2014). To test the impact of clustering on our forecasts, we include models made from both the full and declustered catalogues, assuming that the full catalogues might overestimate the spatial intensity due to observed spatio-temporal clustering and forecast higher rates in areas with recent spatial clustering. We decluster the catalogue by removing events allocated as aftershocks or foreshocks within the UCERF3 catalogue, which were determined by a (Gardner and Knopoff, 1974) clustering algorithm (UCERF3 appendix K). This results in 6 spatial models that we use from this point on, containing components as outlined in Table 1. The posterior mean of the log intensity for each of these models is shown in Figure 3. These models are constructed using an equal-area projection of California and converted to latitude and longitude only in the final step before testing. This figure represents the set of models formed by the training data set.


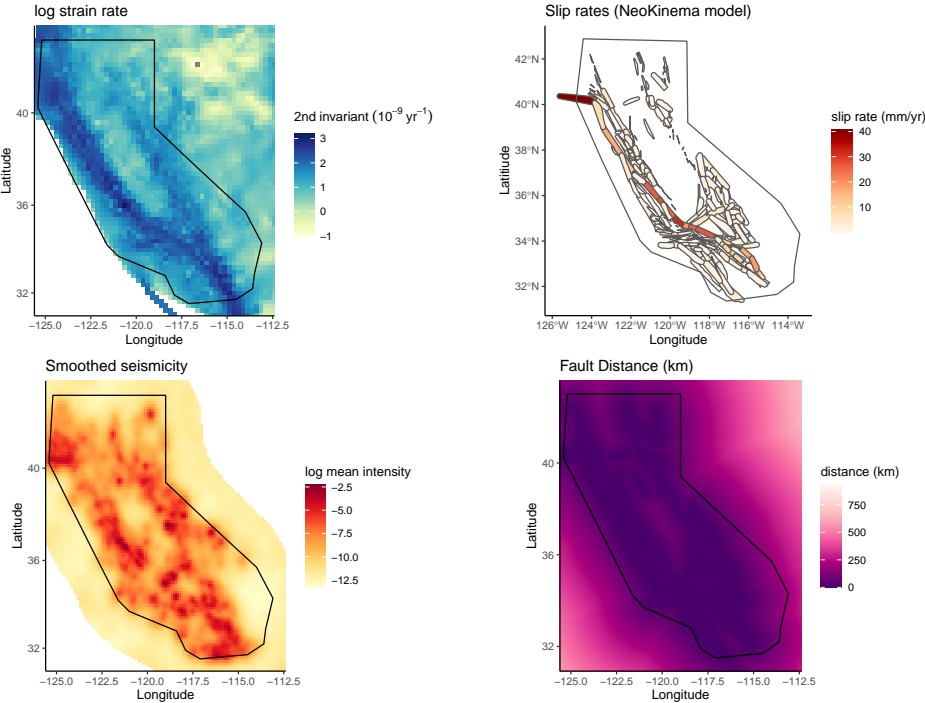

**Figure 2.** Input model covariates (clockwise from top left): GEM strain rate (SR), NeoKinema Slip rates from UCERF3 (NK), distance to nearest (UCERF3, dip and uniformly buffered) fault in km (FD), Smoothed seismicity from a Gaussian random field for events before 1984 (MS).

To extend this approach to a full forecast, we distribute magnitudes across the number of expected events according to a frequency-magnitude distribution. Given the small number of large events in the input training catalogue, a preference between a Tapered Gutenberg-Richter (TGR) or standard Gutenberg-Richter magnitude distribution with a rate parameter $a$, related to the intensity lambda, and an exponent $b$ cannot be fully expressed. The choice of a $b$-value is not straightforward, as the $b$-value can be biased by several factors (Marzocchi et al., 2020) and is known to be affected by declustering (Mizrahi et al., 2021). In this case, we assume $b = 1$ for both clustered and declustered catalogues and for the TGR magnitude distribution we assume a corner magnitude of $M_c = 8$ for the California region proposed by (Bird and Liu, 2007) and used in the Helmstetter et al. (2007) models.

For the gridded forecasts (which assume a uniform event rate or intensity within the area of each square element), we use the posterior mean intensity as shown in Figure 3, transformed to a uniform grid of 0.1 x 0.1 latitude/longitude within the RELM region. We use latitude-longitude here as preferred by the pycsep tests. Magnitudes are then distributed across magnitude bins on a cell-by-cell basis according to the chosen magnitude-frequency distribution and the total rate expected in the cell. In this paper, we show GR magnitudes for the gridded forecasts. For the catalogue-based forecasts, we generate 10,000 samples from the full posteriors of the model components to establish 10,000 realisations of the model spatial intensity within the testing


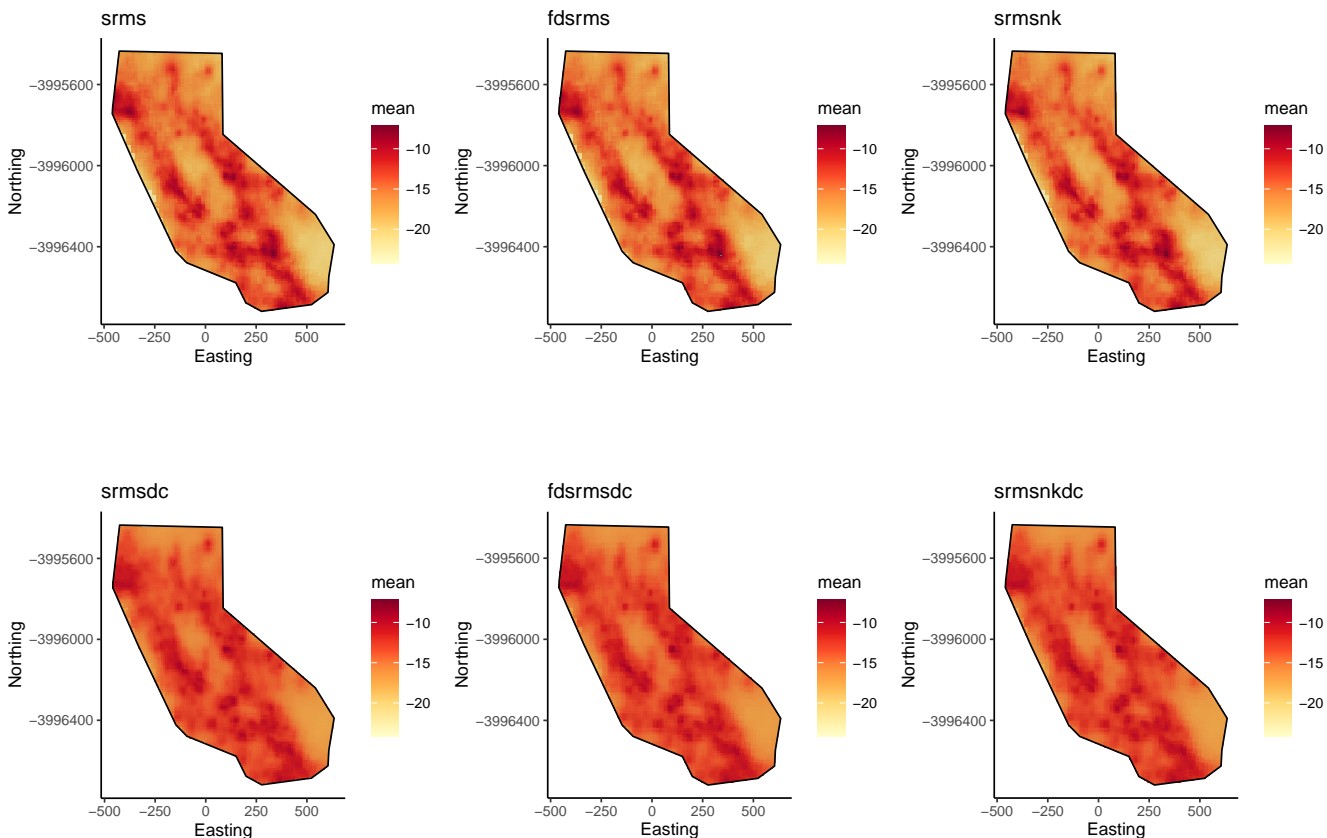

**Figure 3.** Posterior mean intensity for the six inlabru models created with full (top) and declustered (bottom) catalogues of events from 1985-2004.

polygon. We then sample a number of points consistent with the modelled intensity. In this case, we use the expected number of points given the mean intensity (as in step 6 in Figure 1) for one year, and randomly select an exact number of events for a simulated catalogue from a Poisson distribution about the mean rate, scaled to the number of years in the forecast. To sample events in a way that is consistent with modelled spatial rates, we sample many points and calculate the intensity value at the sampled points given the realisation of the model. We then implement a rejection sampler to retain points that have a significantly large intensity ratio compared to the largest intensity in the specific model realisation, with points retained only if the intensity ratio is greater than a uniform random variable between 0 and 1, that is points are retained with probability equal to $1 - \frac{\lambda_p}{\lambda_{max}}$. The set of retained points for each catalogue are then assigned a magnitude sampled from a TGR distribution, by methods described in Vere-Jones et al. (2001). Here we only sample magnuitudes from a TGR distribution in line with the





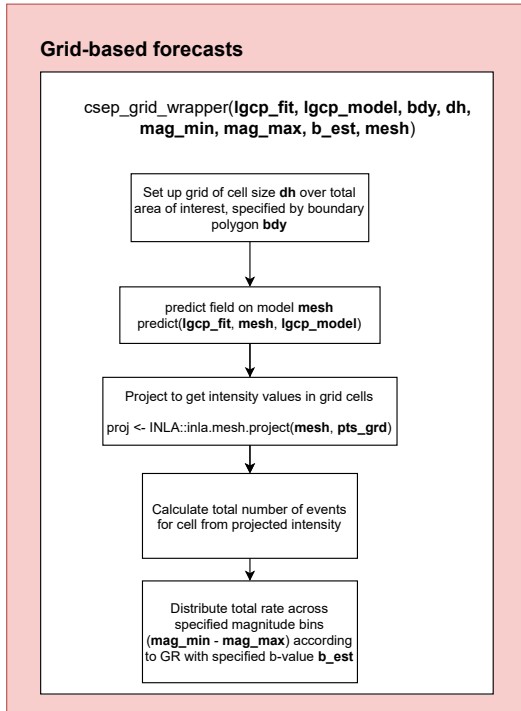

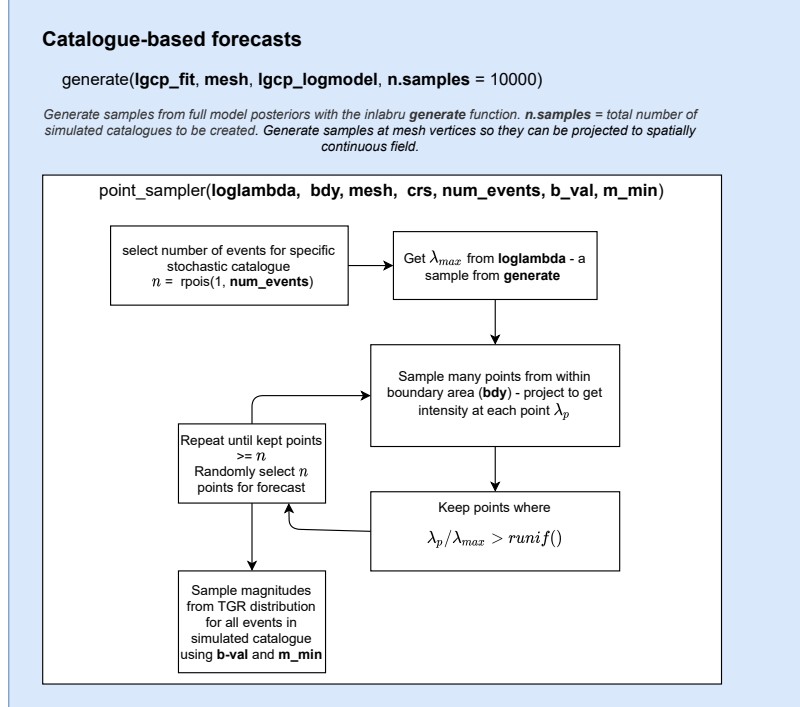

**Figure 4.** Schematic of the code for constructing grid-based (left) and simulated catalogue-based (right) earthquake forecasts given an inlabru LGCP intensity model. These represent step 7 of the workflow.

approach of Helmstetter et al. (2007), to allow a like for like comparison with this benchmark. A schematic diagram showing how grid and catalogue-based approaches are applied is shown in Figure 4.

## 2.2   CSEP tests

To test how well each forecast performs, we first test the consistency of the model forecasts, developed from data between 1984 and 2004, with observations from three subsequent and contiguous 5-year time periods, using standard CSEP tests for
the number, spatial and magnitude distribution and conditional likelihood of each forecast. The original CSEP tests calculate a quantile score for the number (N), likelihood (L) (Schorlemmer et al., 2007) and spatial (S) and magnitude (M) (Zechar et al., 2010) tests, based on simulations that account for uncertainty in the forecast and a comparison of the observed and simulated likelihoods. We use 100 000 simulations of the forecasts to ensure convergence of the test results. The number test is the most straightforward, summing the rates over all forecast bins and comparing this with the total number of observed events. The
quantile score is then the probability of observing at least $N_{obs}$ events given the forecast, assuming a Poisson distribution of the number of events. Zechar et al. (2010) suggests using a modified version of the original N-test that tests the probability of a) at least $N_{obs}$ events with score $\delta_1$ and b) at most $N_{obs}$ events with score $\delta_2$ in order to test the range of events allowed by a forecast. Here we report both N-test quantile scores in line with this suggestion.



The likelihood test compares the performance of individual cells within the forecast. The likelihood of the observation given the model is described by a Poisson likelihood function in each cell and the total joint likelihood described by the product over all bins. The quantile score measures if the joint log-likelihood over many simulations falls within the tail of the observed likelihoods, with the score defined by the fraction of simulated joint log-likelihoods less than or equal to the observed. The conditional likelihood or CL test is a modification of the L-test developed due to the dependence of L-test results on the number of events in a forecast (Werner et al., 2010, 2011). The CL-test normalises the number of events in the simulation stage to the observed number of events in order to limit the effect of a significant mismatch in event number between forecast and observation. The magnitude and spatial tests compare the observed magnitude and spatial distributions by isolating these from the full likelihood. This is again achieved with a simulation approach and by summing and normalising over the other components. For the M-test, the sum is over the spatial bins while the S-test sums over all magnitude bins to isolate the respective components of interest. The final test statistic in both cases is again the fraction of observed log likelihoods within the range of the simulated log likelihood values. In all cases small values are considered inconsistent with the observations - we use a significance value of $0.05$ for the likelihood-based tests and $0.025$ for the number tests to be consistent with previous forecast testing experiments (Zechar et al., 2013).

In the new CSEP tests (Savran et al., 2020), the test distribution is determined from the simulated catalogues rather than a parametric likelihood function. For the N-test the construction of the test distribution is straightforward, being created from the number of events in each simulated catalogue and the quantile score calculated relative to this distribution. For the equivalent to the likelihood test a numerical, grid-based approximation to a point process likelihood is calculated (Savran et al., 2020). This is a more general approach than using the Poisson likelihood as in the grid-based tests, which penalises models that do not conform to a Poisson model. The distribution of pseudo-likelihood is then the collection of calculated pseudo-likelihood results for each simulated catalogue. The spatial and magnitude test distributions are derived from the pseudo-likelihood in a similar fashion to the grid-based approch, as explained in detail by Savran et al. (2020). The quantile scores are calculated similar to the original test cases, but because the simulations are based on the constructed pseudo-likelihood rather than a Poisson likelihood, the simulated-catalogue approach allows for forecasts which are overdispersed relative to a Poisson distribution. Similarly to the original tests, very small values will be considered inconsistent with the observations.

## 3 Full and declustered catalogue models

In constructing the three models both with and without clustering, we can examine relative contributions of the model components given differences in spatial intensity resulting from short-term spatio-temporal clustering. Table 1 shows the posterior mean component of the log intensity for each model both with and without clustering for M4.95+ seismicity, and the number of expected events per year for each model. The greatest contribution in the full-catalogue models comes from the strain rate (SR) for each model, with the past seismicity also making a significant contribution to the intensity. For the models where the catalogue has been declustered, the contribution to the posterior mean from the past seismicity is only slightly lower while the strain rate contribution is much smaller, effectively swapping the relative contributions of these components. This suggests





**Table 1.** Posterior means of model components and number of expected events for full and declustered (DC) models

| Models | mean component contribution to log intensity | | | | |
|---|---|---|---|---|---|
| | strain rate (SR) | past seismicity (MS) | slip rates (NK) | fault distance (FD) | N |
| SRMS | 1.551 | 0.853 | - | - | 6.373 |
| SRMSDC | 0.415 | 0.777 | - | - | 3.679 |
| SRMSNK | 1.488 | 0.837 | 0.017 | - | 6.44 |
| SRMSNKDC | 0.425 | 0.779 | 0.001 | - | 3.79 |
| FDSRMS | 1.574 | 0.857 | - | 0.001 | 6.456 |
| FDSRMSDC | 0.491 | 0.784 | - | 0.004 | 3.737 |

that the strain rate component is more useful when considering the full earthquake catalogue than when the catalogue has been declustered. In both full- and declustered-catalogue models, the number of expected events is similar across all three models, thus we expect the models to perform similarly in the CSEP N-tests.

Figure 3 shows that the declustered-catalogue models (bottom row) appear much smoother than those constructed from the full catalogue, as they have lower intensity in areas with large seismic sequences in the training period. They also have a smaller range in intensity than the full catalogue models, with the (median) highest rates lower and the (median) lowest rates higher than the full catalogue models, meaning they cover less of the extremes at either end.

## 4  Model testing

We now test the models using the pyCSEP package for python (Savran et al., 2021). We begin with the standard (grid-based) CSEP test models described by Schorlemmer et al. (2007); Zechar et al. (2010) included in pyCSEP and described in section 2.2.

### 4.1  Grid-based forecast tests

We first compare the performance of our five-year forecasts, developed with a training window of 1984-2004, over the testing period 01/01/2006-01/01/2011 with the Helmstetter et al. (2007) forecast. In this time, the comcat catalog (https://earthquake.usgs.gov/data/comcat/). includes 32 M4.95+ events in the study region defined by the RELM polygon. All the models, regardless of their components or which catalogue is used, perform well in the magnitude tests due to the use of the GR distribution. The forecast tests are shown visually in Figure 5 and the quantile scores are reported in Table 2 for all tests and time-periods. A model is considered to pass a test if the quantile score is $\geq 0.05$ for all tests except the N-test, where the significance level is set at $\geq 0.025$ for both score components and the model fails if either score fails (Schorlemmer et al., 2010; Zechar et al., 2010). In Figure 4 the observed likelihood is shown as a coloured symbol (red circle for a failed test and green square for a passed one) and the forecast range is shown as a horizontal bar, for ease of comparison. In the number test (N-test), the declustered forecasts underpredict the number of expected events significantly in all cases due to the much smaller





**Table 2.** Quantile scores for CSEP tests. Upper bounds for S, L and PL-tests, lower bound for N. Bold indicates consistency with observations, italics highlight declustered models.

| Time | Models | Gridded | | | | | Catalogue | | | | |
|---|---|---|---|---|---|---|---|---|---|---|---|
| | | N-test ($\delta_1$) | N-test ($\delta_2$) | S-test | M-test | CL-test | N-test ($\delta_1$) | N-test ($\delta_2$) | S-test | M-test | PL-test |
| 2006 - 2011 | SRMS | **0.465** | **0.603** | 0.025 | **0.288** | **0.105** | **0.440** | **0.625** | **0.180** | **0.596** | **0.268** |
| | *SRMSDC* | *0.002* | *0.999* | *0.738* | *0.290* | *0.848* | *0.002* | *0.999* | *0.922* | *0.158* | *0.006* |
| | FDSRMS | **0.463** | **0.605** | 0.032 | **0.289** | **0.122** | **0.462** | **0.606** | **0.196** | **0.617** | **0.305** |
| | *FDSRMSDC* | *0.002* | *0.999* | *0.692* | *0.291* | *0.818* | *0.001* | *0.999* | *0.891* | *0.162* | *0.007* |
| | SRMSNK | **0.486** | **0.583** | 0.028 | **0.293** | **0.115** | **0.463** | **0.605** | **0.243** | **0.611** | **0.327** |
| | *SRMSNKDC* | *0.002* | *0.999* | *0.711* | *0.288* | *0.830* | *0.002* | *0.999* | *0.874* | *0.167* | *0.007* |
| 2011 - 2016 | SRMS | 0.999 | 0 | 0.039 | **0.158** | 0.026 | 1 | 0 | 0.011 | **1** | **0.999** |
| | *SRMSDC* | *0.963* | *0.064* | *0.766* | *0.153* | *0.485* | *0.952* | *0.081* | *0.807* | *0.963* | *0.951* |
| | FDSRMS | 0.9999 | 0 | 0.030 | **0.155** | 0.021 | 1 | 0 | 0.011 | **1** | **0.999** |
| | *FDSRMSDC* | *0.964* | *0.063* | *0.744* | *0.156* | *0.467* | *0.958* | *0.070* | *0.883* | *0.965* | *0.960* |
| | SRMSNK | 0.999 | 0 | **0.070** | **0.157** | **0.044** | 0.999 | 0 | 0.0132 | **0.999** | **0.9999** |
| | *SRMSNKDC* | *0.960* | *0.068* | *0.766* | *0.158* | *0.487* | *0.962* | *0.063* | *0.793* | *0.968* | *0.959* |
| 2016 - 2021 | SRMS | **0.792** | **0.264** | 0 | **0.369** | 0.002 | **0.772** | **0.283** | 0.003 | **0.564** | **0.312** |
| | *SRMSDC* | *0.029* | *0.982* | *0.008* | *0.368* | *0.068* | *0.019* | *0.989* | *0.081* | *0.141* | *0.005* |
| | FDSRMS | **0.791** | **0.266** | 0 | **0.367** | 0.004 | **0.795** | **0.268** | 0.005 | **0.602** | **0.369** |
| | *FDSRMSDC* | *0.030* | *0.981* | *0.0068* | *0.367* | *0.062* | *0.025* | *0.986* | *0.073* | *0.149* | 0.007 |
| | SRMSNK | **0.806** | **0.248** | 0 | **0.367** | 0.006 | **0.789** | **0.266** | 0.006 | **0.587** | **0.374** |
| | *SRMSNKDC* | *0.027* | *0.984* | 0.005 | *0.369* | *0.050* | *0.027* | *0.983* | *0.119* | *0.151* | 0.009 |

number of expected events per year and the large number of events that actually occurred in the testing time period. In spatial
testing (S-test), the full-catalogue models all perform poorly. In contrast, the declustered catalogue models all pass the S-test.
In the conditional likelihood tests (CL-test), all of the models perform well and pass the CL-test (figure 5), with the declustered
models performing better due to better spatial performance.

We then repeat the tests for two additional five year periods of California earthquakes illustrated in Figure 5. In all time
windows, the M-test results remain consistent across all models. In the 2011-2016 period, there are 13 M4.95+ events within
the RELM polygon, and this significant reduction in event number means that our full-catalogue models and the Helmstetter
models all overestimate the actual number of events significantly, with the true number outwith the 95% confidence intervals of
the models. In contrast, most of the models perform better in the S-test during this time period with the full catalogue slip-rate
model and all declustered-catalogue models recording a passing quantile score (Table 2). Each of the models made with a
declustered catalogue passes the CL-test.
In the 2016-2021 period (Figure 5 top) there are 30 M4.95+ events, which is within the confidence intervals shown for all
tested models so all models pass the N-test for the first time. However none of the tested models pass the S-test due to the



**Figure 5.** Grid-based forecast tests for all forecasts for three five year time periods: 2006-2011 (top), 2011-2016 (middle) and 2016-2021(bottom). The bars represent the 95% confidence interval derived from simulated likelihoods from the forecast, while the symbol represents the observed likelihood for observed events. The green square identifies that a model has passed the test and a red circle indicates inconsistency between forecast and observation. The forecasts are compared to both the full (Helmstetter aftershock) and declustered models of Helmstetter et al (2007)

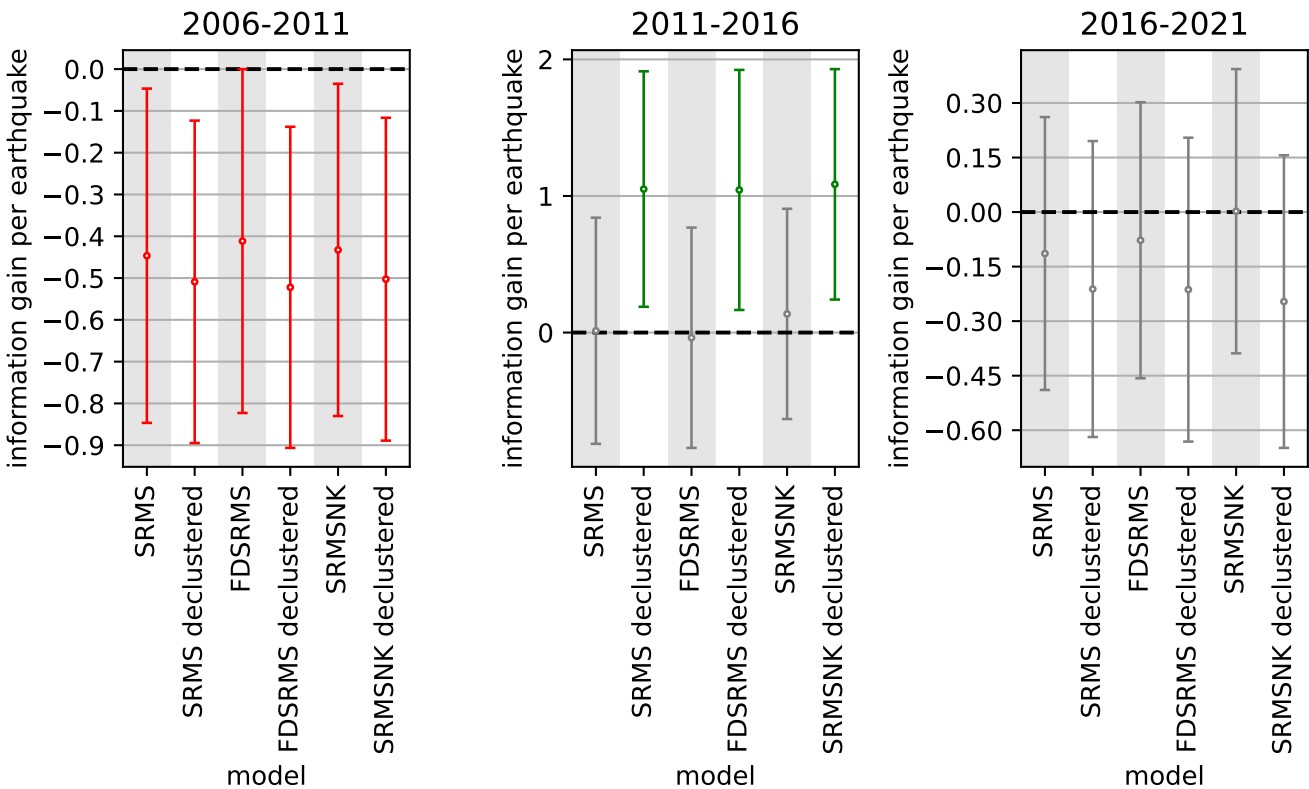

**Figure 6.** T-test results for the inlabru models showing information gain per earthquake relative to the full Helmstetter et al (2007) model (helmstetter aftershock in Fig. 5) for three time periods. Red indicates forecasts are worse in terms of information gain and green indicates forecasts performing better than the benchmark forecast. Grey forecasts are not significantly different in terms of information gain.

spatial distribution of the events in this time period being highly clustered in areas without exceptionally high rates, even for models developed from the full catalogue. The CL-test results for the 2016-2021 period show that none of the models perform particularly well in this time period, with two of the declustered-catalogue models passing the test only barely.

These statistical tests (N, S, M and CL) investigate the consistency of a forecast made during the training window with the observed outcome. They do not compare the performance of models directly with each other, but rather with observed events. One method of comparing forecasts is by considering their information gain relative to a fixed model with a paired T-test (Rhoades et al., 2011). Here, we implement the paired T-test for the gridded forecast to test their performance against the Helmstetter et al. (2007) aftershock forecast as a benchmark, because it performed best in comparison to other RELM models

in previous CSEP testing over various timescales (Strader et al., 2017). The results of the comparison are shown in Figure 6. For the first time period (2006-2011), the models perform similarly in terms of information gain, and all of the inlabru models perform worse than the Helmstetter model. For the 2011-2016 period, the inlabru models developed from the declustered catalogues perform better in terms of information gain than those developed from the full catalogue and significantly better than





the Helmstetter model. In the most recent testing period (2016-2021), the inlabru models have an information gain range that
includes the Helmstetter model. Together these results imply the inlabru models provide a positive and significant information
gain on a 5-10 year time period after the end of the training period for declustered-catalogue models, and not otherwise.

### 4.2 Simulated-catalogue forecasts

Our second stage of testing uses simulated catalogues in order to make use of the newer CSEP tests (Savran et al., 2020). We
use the number, spatial and pseudolikelihood (PL) tests to evaluate these forecasts, with the PL test replacing the grid-based
L-test. In our case, as described above the number of events in the simulated catalogues is inherently Poisson due to the way
they are constructed, but the spatial distribution is perturbed from a homogeneous Poisson distribution due to the contributions
of model covariates and the random field itself (e.g. see equation 1, where a homogenous Poisson process would include only
the intercept term $\beta_0$) and the parameter values are sampled from the posterior at each simulation, so vary from simulation to
simulation. Figure 7 shows the test distributions for each forecast as a letter-value plot (Hofmann et al., 2011), an extended
boxplot which includes more quantiles of the distribution until the quantiles become too uncertain to discriminate. This allows
us to understand more of the full distribution of model pseudo-likelihood than a standard quantile range or boxplot, while
allowing easy comparisons between the results for different forecast models.

We expect the grid-based and simulated-catalogue approaches to have similar results in terms of the magnitude (M) tests
due to the similarity of magnitude distributions used in construction, and all models do similarly well in this test (Table 2).
Similarly, we do not expect significant differences in the number tests with this approach, since our method of determining the
number of events will result in a Poisson distribution of the number of events. However, since the number of events varies in
each synthetic catalogue we can look at the distribution of the number of events in the synthetic data produced by the ensemble
of forecast catalogs relative to the observed number. This is shown in the left panel of Figure 7, with the observed number of
events for each time period shown with a dashed line. Again, the declustered models do better in the 2011-2016 period, though
it is clear the observed number of events is low even for them.

We might expect the most noticeable differences to occur in the spatial test, because it measures the spatial component
consistency with observed events and because we are now using the full posterior distribution of spatial components, and
therefore potentially allowing more variation in the observed spatial models. The middle panel of Figure 7 shows the spatial
likelihood distribution constructed from simulated catalogues.
Similar to the grid-based examples, for the 2006-2011 period (red star indicator) the spatial performance of the SRMS and
FDSRMS models is better when the full, rather than declustered catalogue, has been used in model construction.

All of the models pass the S-test when considering quantile scores in this time period. Similarly, when testing the 2011-2016
period (test statistic shown with a blue diamond), all of the models built from the declustered catalogue pass the S-test, while
the full-catalogue models do more poorly. In 2016-2021 (green circle), the spatial performance of all models is again poor. The
best-performing model in this time period is the FDSRMS-declustered model (Table 2), with the declustered-catalogue models
generally doing better than the full-catalogue models.

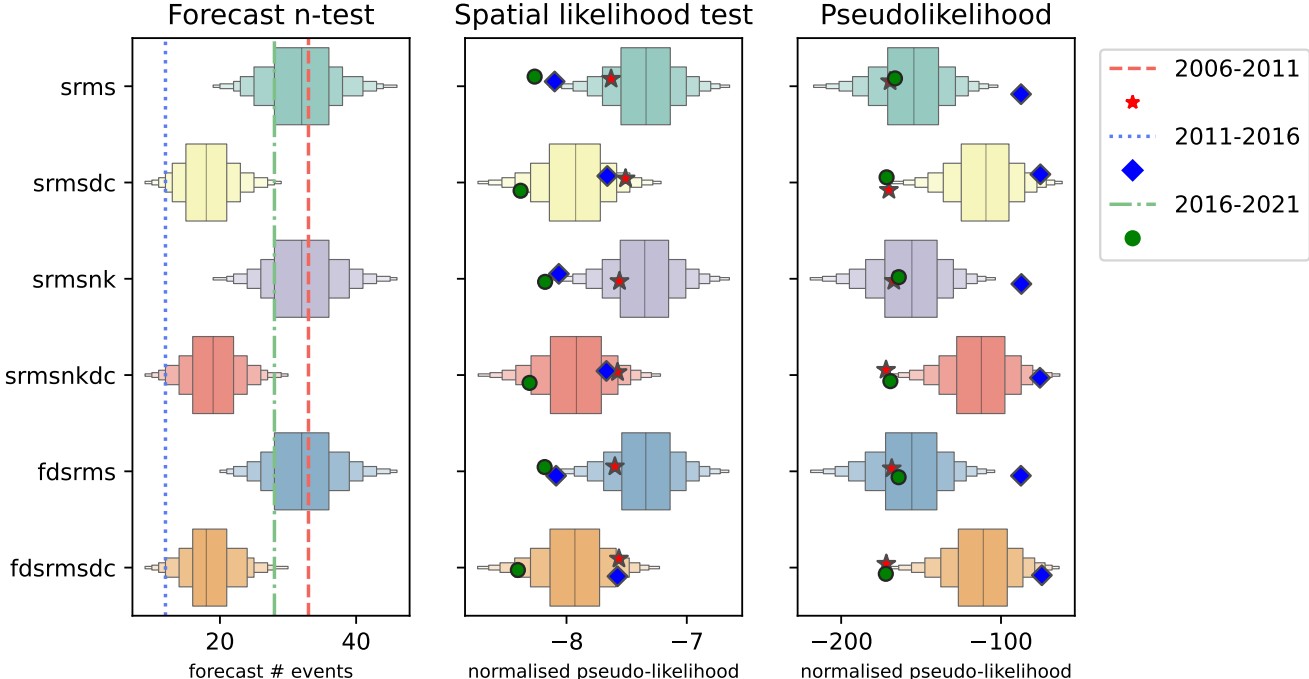

**Figure 7.** N-test, S-test and pseudo-likelihood results for each of the 6 inlabru models when forecasts are generated from 10 000 synthetic catalogues sampling from the full inlabru model posteriors. For the n-test, the number of observed events for the 2006-2011, 2011-2016 and 2016-2021 are shown by the red, blue and green dashed lines respectively. For the S- and Pseudo-likelihood tests, the observed test statistic for each time period is shown as a symbol (red star for 2006-2011, blue diamond for 2011-2016 and green circle for 2016-2021)

Finally, the pseudo-likelihood test (Figure 7, right) incorporates both spatial and rate components of the forecast, much like the grid-based likelihood. For the inlabru models, the preference between the models for the full and declustered catalogues changes with time period with both sets of models doing poorly in the 2016-2021 period (green circle). All of the full-catalogue

models pass in 2006-2011 and surprisingly in 2011-2016 and 2016-2021, though some of the quantile scores are again quite large and in the upper tails of the likelihood distributions. Like the grid-based likelihood test, the pseudo-likelihood test penalises for the number of events in the forecast, which allows the full-catalogue models to pass the pseudo-likelihood test even when they have poor spatial performance, as in the 2011-2016 and 2016-2021 testing periods.

## 5    Discussion

### 5.1    Number of events

While the full-catalogue models performed well in the tests for the first five-year time window, the other two sets of test results were less promising. This can be largely explained by the number of events that occurred in the 10 year period from 2006-2016

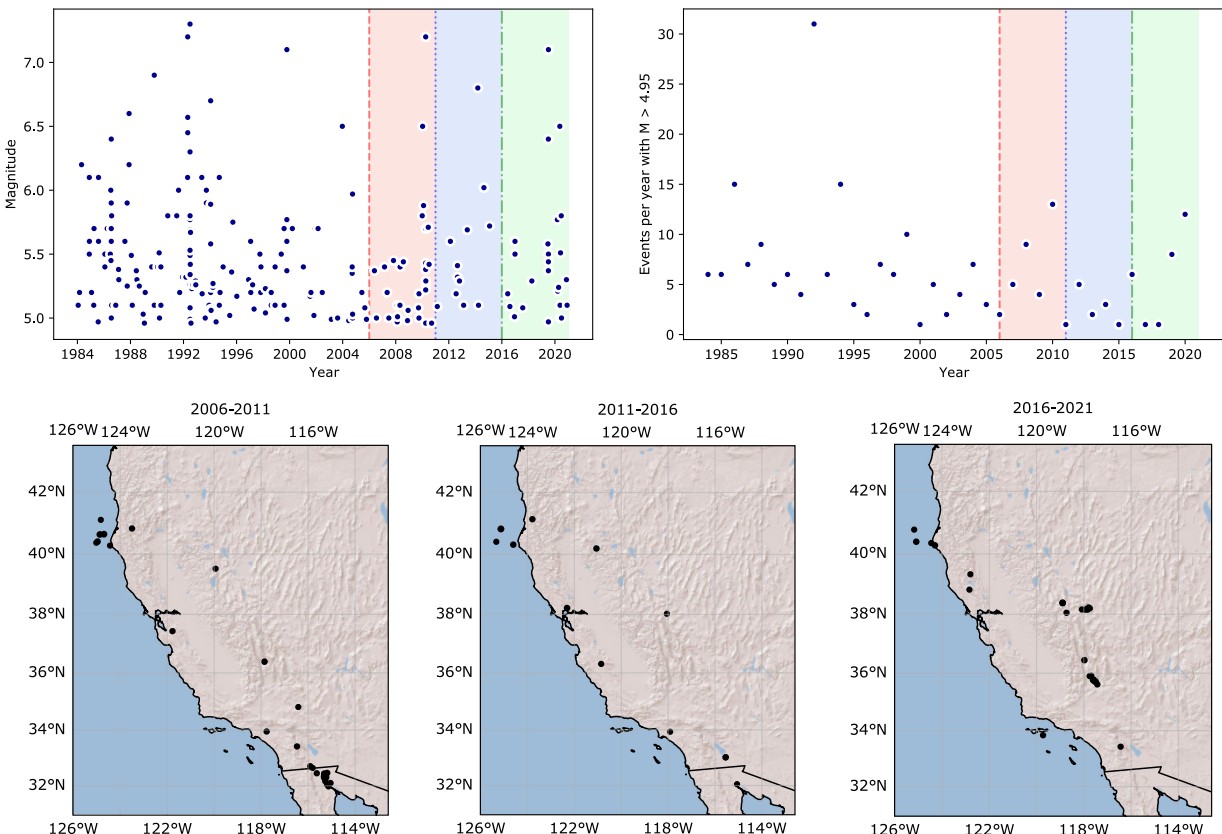

**Figure 8.** Top: Catalogue of events in California from 1985-2021. The period 1984-2004 is used for model construction, and the three testing periods are shown with red, blue and green backgrounds. The left panel shows the magnitude of events in time and the right the number of events in each year. Bottom: the comcat catalogues for the three five-year testing intervals.

(red and blue backgrounds in Figure 8, top right). In this time 45 events were recorded in the comcat catalog, compared to 32 events in the five years between 2006-2011. In the twenty years from 1984-2004 used in our model construction, a total of 156 events with $M > 4.95$ were recorded, which is an average of 7.8 events/year. Helmstetter et al. (2007) explicitly use the average number events per year with magnitude $> 4.95$ (7.38) to condition their models. It is therefore not surprising that the declustered forecasts perform oppositely, with poor performance in the 2006-2011 time period and better performances in the 2011-2016 time period when fewer events occurred. This is a common issue in CSEP testing, reported both in Italy when the five-year tests occurred in a time period with a large cluster of events in a historically low-seismicity area (Taroni et al., 2018) and in New Zealand, where the Canterbury earthquake sequence occurred in the middle of the CSEP testing period (Rhoades et al., 2018) resulting in significantly more events than expected. Strader et al. (2017) found that four of the original RELM forecasts overpredicted the number of events in the 2006-2011 time window and 11 overpredicted the number of events in the



second 5-year testing window (2011-2016), including the Helmstetter model. Overall, the inlabru model N-test results were comparable to the Helmstetter model performance in the grid-based assessment and performed well at forecasting at least the minimum number of events in all but the declustered models in the first testing period (table 2).

## 5.2 Full- and declustered-catalogue models

We did not filter for mainshocks in the observed events, so we might expect the N-test results for the declustered models to do poorly, but they were consistent with observed behaviour in 2 of the 3 tested time periods in both the grid-based and catalogue testing. If we consider only the lower bound of the N-test, the declustered models pass the test in the full 2011-2021 time period and only perform poorly in 2006-2011, a time period which arguably contained many more than average events (Figure 8). Similarly, the full catalogue models do poorly on the upper N-test in 2011-16 but otherwise pass in time windows with higher numbers of events.

The declustered models pass spatial tests more often than the full catalogue models because they are less affected by recent clustering, and perhaps benefit from being smoother overall than the full-catalogue models (Figure 3). The superior performance of the declustered models may not have been entirely obvious had we tested only the 2006-2011 period and relied solely on the 'pass' criterion from the full suite of tests: only the full-catalogue synthetic catalogue forecast models get a pass in all consistency tests in this time period. This highlights a need for forecast to be assessed over different timescales in order to truly understand how well they perform, a point previously raised by Strader et al. (2017) when assessing the RELM forecasts, and more generally embedded in the evaluation of forecasting power since the early calculations of Lorenz (1963) for a simple but nonlinear model for Earth's atmosphere in meteorological forecasting. .

We conclude that neither a full nor declustered catalogue necessarily gives a better estimate of the future number of events in any 5-year time-period, though the declustered models tend to perform better spatially, and may be more suitable for longer-term forecasting. Given different declustering methods may retain different specific events and different total numbers of events, different declustering approaches may lead to significant differences in model performances, especially in time periods with a small number of events in the full catalogue. To truly discriminate between which approach is best, a much longer testing time frame would be needed to ensure a suitably large number of events.

## 5.3 Spatial performance of gridded and simulated catalogue forecasts

In general, the simulated catalogue-based forecasts were more likely to pass the tests than the gridded models. This is most obvious in the first testing period, when the simulated catalogue-based models based on the full-catalogue passed all tests and those for the declustered catalogues only fail due to the smaller expected number of events. Similarly, in the most recent testing period (2016-2021) the simulated-catalogue forecasts are able to just pass the S-test where all models fail in the gridded approach.

The simulated catalogue approach allows us to consider more aspects of the uncertainty in our model. For example, we could further improve upon this by considering potential variation in the b-value in the ensemble catalogues which arises from





magnitude uncertainties, an issue that may be particularly relevant when dealing with homogenised earthquake catalogues (Griffin et al., 2020) or where the b-value of the catalogue is more uncertain (Herrmann and Marzocchi, 2020).

## 5.4 Roadmap - where next?

The main limitation of the work presented here, and many other forecast methodologies, is how aftershock events are handled. Our choice of (a relatively high) magnitude threshold for modelling may have also benefited the full model by ignoring many

small magnitude events that would be removed by a formal declustering procedure. The real solution to this is to formally model the clustering process.

    The approach presented here conforms strongly with current practice. In time-independent forecasting and PSHA, catalogues are routinely declustered to be consistent with Poisson occurrence assumptions. Operational forecasting already relies heavily on models such as the epidemic type aftershock sequence model(ETAS, Ogata (1988)) to handle aftershock clustering (Mar-

zocchi et al., 2014), but few attempts have been made to account for background spatial effects beyond a simple continuous Poisson rate. The exceptions to this are changes to the spatial components of ETAS models (Bach and Hainzl, 2012), the recent developments in spatially-varying ETAS (Nandan et al., 2017) and extensions to the ETAS model that also incorporate spatial covariates (Adelfio and Chiodi, 2020). However, the more versatile inlabru approach allows for more complex spatial models than has yet been implemented with these approaches. The inlabru approach also provides a general framework to test the

importance of different covariates in the model, and a fully Bayesian method for forecast generation as we have implemented here.

    One way to handle these conflicts is to model the seismicity formally as a Hawkes process, where the uncertainty in the tradeoff between the background and clustered components is explicit and can be formally accounted for. In future work we will modify the workflow of Figure 1 to test the hypothesis that this approach will improve the ability for inlabru to forecast

using both time-independent and time-dependent models.

## 6   Conclusions

For the first time, we present time independent forecasts for California developed with inlabru. We developed three earthquake forecasts for California considering different combinations of spatial covariates and developed with both the full and declustered catalogue in each case, resulting in 6 models in total. These models each include spatial covariates that perform well in

retrospective testing of spatial seismicity, which are then extended to spatio-temporal models by considering the frequency-magnitude distribution and assuming a Poisson distribution of events in time. The full-catalogue models each pass the standard CSEP tests for number, magnitude and spatial distribution, and perform favourably with the Helmstetter model tested in the original RELM experiment over the 2006-2011 period, demonstrating the suitability of inlabru models for time-independent earthquake forecasting. The declustered catalogues perform less well in this time period due to the lower expected number

of events, but perform better in spatial tests and overall in the 2011-2016 time period, where the full catalogue models over-estimate the number of events quite significantly. Neither the full-catalogue or declustered-catalogue models perform well in





the most recent testing period, with much worse spatial performance. Simulated catalogue forecasts that make use of the full posteriors of the model pass consistency tests more often than their grid-based equivalents by better accounting for uncertainty in the model itself.

*Code and data availability.* The code and data required to produce all of the results in this paper, including figures, can be downloaded from https://doi.org/10.5281/zenodo.5793157

*Author contributions.* Kirsty Bayliss developed the methodology, carried out the formal analysis and interpretation, and wrote the first draft of the paper. Farnaz Kamranzad contributed significantly to visualisation, particularly development of Figure 1. Mark Naylor and Ian Main contributed to the conceptual design, the interpretation of the results, and the writing of the paper. All authors contributed to paper review
and drafting.

*Competing interests.* The authors declare that they have no conflict of interest.

*Acknowledgements.* This work is funded by the Real-time Earthquake Risk Reduction for a Resilient Europe 'RISE' project, which has received funding from the European Union's Horizon 2020 research and innovation programme under grant agreement No 821115. We thank Francesco Serafini and Finn Lindgren for helpful discussions and suggestions.



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
