# Peer review of "Pseudo-prospective testing of 5-year earthquake forecasts for California using inlabru"

_Natural Hazards and Earth System Sciences, 2021_

## Author Comment (AC1)

[Figure]

Figure 1: Pairwise comparison of models for full catalogue models. The top-right side of the plot shows differences in log median intensity and the lower left section shows the differences in model variances between the different models. The median log intensities for each model are shown on the diagonal. Models include combinations of smoothed past seismicity (MS), strain rate (SR), fault distance (FD) and fault slip rates (NK).

[Figure]

Figure 2: Pairwise comparison of models for declustered catalogue models. The top-right side of the plot shows differences in log median intensity and the lower left section shows the differences in model variances between the different models. The median log intensities for each model are shown on the diagonal. Models include combinations of smoothed past seismicity (MS), strain rate (SR), fault distance (FD) and fault slip rates (NK).

---

## Author Response (AR1)

We thank the reviewers for their time and thoughtful comments on our article. Below, we address each of their comments, with reviewer's comments highlighted in blue italics and our responses underlined and in black.

*Reviewer 1:*

*The paper is generally well written and sound hence it is suitable for publication with minimal changes.*

*At a first read is very hard to follow because it continuously refers to the companion paper by Bayliss et al. (2020). The authors should describe some more details of the forecasting models (e.g. SRMS, SRMSDC etc.) and of testing methods (e.g. DIC) defined in Bayliss et al. (2020) so that the reader is not obliged (as I had to do) to read the latter paper to understand the present one.*

We have tried to strike a balance between our previous work and the extensions addressed in this paper, and we apologise that we have not done this well enough. We have added clearer descriptions of the forecast models and expanded upon our use of DIC as a testing method for model discrimination as an appropriate log-relative likelihood metric that fairly penalises models with larger numbers of parameters.

*I do not think that Fig. 1 and 4 are particularly useful, and then can be omitted without loss of information.*

Figure 1 aimed to summarise necessary steps in model construction without too much repetition of previous work, and, on testing with potential users of the method (primarily research students), we feel it is useful for explaining the modelling process, and to ensure reproducibility of our results by independent researchers. This is now explained in the text.

Likewise, Figure 4 aimed to provide a summary of the steps of forecast creation that may be helpful to other researchers. With both models, our aim was to facilitate reproducibility by other researchers, and we have clarified this in the text.

*Panels in Fig. 3 can hardly be distinguished one to the other. Try changing somehow the color palette.*

This is a very fair point, however the similarity in the models is the issue, rather than any particular choice of palette: the models are very similar to the eye, especially when we are only considering the median values.

To help make the differences between these models clearer, we have added new figures for the full-catalogue and declustered-catalogue models (Figures 2 and 3) which are plots showing pairwise differences between the log median intensity values (top-right) and variances (bottom-left). This better highlights differences in the intensity models used for the forecasts.

*Line 211. Figure 5 instead of Figure 4*

Thank you for pointing this out, we have updated the figure number accordingly.

*The conclusions are weak, please try to better explain what you have learned from this work.*

This is also a fair point on re-reading the text. We have amended the conclusion to better explain the key findings of the work to a general audience.

Reviewer 2:

*In this paper, the authors applied the open-source inlabru method to time-independent earthquake forecasts. They used the California region, defined for the RELM experiment, as a case study. The authors described the methodology details in another scientific paper just published. In the first part of the paper, the authors described: i) the spatial models applied in broad terms, ii) the gridded forecast and the synthetic catalog obtained from the method application, and iii) the test applied for the validation of results. For the model construction, the authors examined the relative contributions of the full and declustered catalogs. In the second part, the authors analyze the results obtained with the proposed methodology applying both grid-based and synthetic catalog tests included in the PyCSEP system. Furthermore, they compared the performance obtained with their models with that produced by Helmstetter in 2006 and submitted in the RELM experiment. They concluded that: (i) the full-catalog models performed well in retrospective testing (number, magnitude, and spatial distribution) for the first period 2006-2011 and the results are comparable with those produced by the Helmstetter model; (ii) for the period 2011-2016 the declustered catalog models performed better than the full catalog models, (iii) in the period 2016-2021 the models performed better the N-Test respect to S-Test and the CL-Test and (iv) the simulated catalogs forecasts pass the consistency test more often than their grid-based forecasts.*

*The paper is satisfactory, and the methodological approach is partially described in the text and referred to another published paper. The data and the code are available for free. The article is well written, and it represents a development in the integration of data from different sources. The use of tests that incorporate grids and synthetic seismic catalogs is also appreciable.*

*I suggest that the paper must be published after minor revision.*

*I recommend only a few comments about the paper:*

    1. *Why don't you use the 2005 data as input or in the testing phase?*

The testing phase was chosen to be 2006-2011 to be directly comparable with the original CSEP/RELM testing periods, in particular to allow a direct comparison with the results of Helmstetter et al (2007), which forms a benchmark as the most successful model in that first test. The input data was originally chosen to be 1984-2004 in line with our previous work. There was no intention to miss out 2005 data in both phases.

Nevertheless, we have updated the results so that models are trained on the 1985-2005 time period to avoid any confusion for the reader, though forecasts for both training periods will continue to be available through github and Zenodo. Updating to the new training period does not significantly affect the results, with minor changes in spatial performance observed in the first testing period. We have added results from the 2004-1984 data as supplementary material.

    2. *The imposition of b = 1 for the declusterized catalog probably affected the results obtained. When the catalog is declustered, b tends to be lower than one due to the lack of smaller events. What were the real values of the b-value in the complete and declustered real catalog?*

We agree that the choice of b-value for the declustered forecasts may not have been ideal, though we would argue it is not a bad null hypothesis to start with. Given the small size of the catalogues, there is likely to be significant uncertainty in any single b-value used in this way.

While this choice may have had implications for the magnitude test, it should not have adversely affected other tests, and in fact the magnitude test results are acceptable in two out of three time windows despite this potential flaw. We have clarified this in the model description and testing sections of the paper and will investigate this more fully in future work.

While investigating this, we realised that the magnitude test results were incorrect, reporting the wrong quantile scores. We have now updated this in the results section, which results in a better performance for the declustered catalogues in the magnitude tests and a poorer performance for all models in the 2011-2016 testing period. We also updated the 1984-2004 results in the supplementary material to correct for this.

*3. In the tests, you have combined various input data that you had in your possession. Why didn't you test the model with only past seismicity? In this way, it was possible to see how other data contributed to the result. It might also be interesting to see the seismic catalog alone in the synthetic catalog test.*

In order to keep to a limited number of forecasts, we chose spatial models that performed best at describing California seismicity according to their DIC in Bayliss et al 2020 to develop to full time-independent forecasts. In this regard, the past seismicity input only is the second-worst performing model in that paper. We chose instead to compare our models to the Helmstetter et al (2007) past seismicity model rather than our own because of the better performance of the Helmstetter model in the RELM tests, and a desire to keep the number of different models to a reasonable size.

*4. Figures 3, 5, 6, and 7: insert in the caption the various acronyms (MS, SR, FD, NK, and DC) to facilitate the reader to understand the results;*

The figure captions have been updated to include a description of the acronyms.

*5. Page 10, line 211: change "Figure 4" to "Figure 5".*

Thank you for pointing this out, we have updated the figure number accordingly.